# LLaVA-Critic:
# Learning to Evaluate Multimodal Models

## Abstract

We introduce LLaVA-Critic, the first open-source large multimodal model (LMM) designed as a generalist evaluator to assess performance across a wide range of multimodal tasks. LLaVA-Critic is trained using a high-quality critic instruction-following dataset that incorporates diverse evaluation criteria and scenarios. Our experiments demonstrate the model's effectiveness in two key areas: $(i)$ LMM-as-a-Judge, where LLaVA-Critic provides reliable evaluation scores, performing on par with or surpassing GPT models on multiple evaluation benchmarks; and $(ii)$ Preference Learning, where it generates reward signals for preference learning, enhancing model alignment capabilities. This work underscores the potential of open-source LMMs in self-critique and evaluation, setting the stage for future research into scalable, superhuman alignment feedback mechanisms for LMMs.

## 1 Introduction

The ability of learning to evaluate is increasingly taking on a pivotal role in the development of modern large multimodal models (LMMs), as pre-training on existing web data reaches maturity and the focus is shifting towards post-training with AI-enhanced synthetic data, which shows growing potential. Reliable AI evaluation is essential, not only for offering scalable solutions to reduce human labor in complex task assessments, but also for generating effective reward signals in reinforcement learning and guiding inference-time search (Ouyang et al., 2022; OpenAI, 2024a; Snell et al., 2024).

Existing LMMs have made tremendous progress in recent research community (Li et al., 2023a), primarily on improving the performance of various real-world vision tasks in single-image (Liu et al., 2023b; Bai et al., 2023; Chen et al., 2023b), multi-image (Li et al., 2024c; Jiang et al., 2024) and video scenarios (Li et al., 2024b; Lin et al., 2024; Wang et al., 2024b). It remains unexplored to develop open LMMs to play the role of a judge and evaluate the performance of multimodal models. For instance, a model can follow a well-designed, itemized evaluation criterion to provide a score between 1 and 10 for rating different model responses in a visual chat task (Liu et al., 2023b). Along with the score, it would also offer the associated reasoning behind the evaluation, ensuring transparency and consistency in assessing model performance. In this paper, we present the first attempt to curate the instruction-following data particularly for evaluation, based on which we develop a LMM, LLaVA-Critic. Two primary scenarios/goals of building LLaVA-Critic are highlighted:

- *Scenario 1: LMM-as-a-Judge*. Open-source LMMs that can deliver reliable evaluation scores, comparable to or surpassing proprietary models like GPT-4V (OpenAI, 2023)/GPT-4o (OpenAI, 2024b). These models can serve as a free alternative to replace commercial GPT models in various evaluation benchmarks.
- *Scenario 2: Preference Learning*. A scalable solution for generating effective reward signals, reducing the need for costly human feedback collection. This approach enhances preference alignment with AI-generated feedback.

Our experimental results demonstrate that: $(i)$ As a judge model, the evaluation scores and rankings provided by LLaVA-Critic show a high correlation with commercial GPT models, making it a cost-effective alternative for model developers in resource-constrained settings; $(ii)$ In preference learning, LLaVA-Critic offers AI-generated feedback in iterative Direct Preference Optimization (DPO) (Rafailov et al., 2024), outperforming the preference signals provided by the reward model in LLaVA-RLHF (Sun et al., 2023), which relies on human feedback for training the reward model.

In summary, our contributions are as follows:

- *Critic Instruction-Following Data*: We present a high-quality dataset tailored to follow instructions in complex evaluation setting to provide quantitative judgment and the corresponding reasoning process. It consists of 46k images with 113k evaluation instruction samples, including both pointwise and pairwise evaluation settings.
- *Large Multimodal Models*: We develop LLaVA-Critic, LMMs that expand the capabilities of open models to play of the critic, providing effective evaluation and feedback.
- *Open-Source*: In an effort to support the development of general-purpose visual assistants, we release our critic instruction data, codebase, model checkpoints, and the trained visual chat demo to the public.

## 2 RELATED WORK

**LMM-as-a-judge.** Strong proprietary LMMs such as GPT-4V / GPT-4o have been demonstrated to serve as generalist evaluators for vision-language tasks (Zhang et al., 2023a; Ge et al., 2023). Specifically, for complex scenarios related to human judgment, such as visual chat and detailed captioning, LMMs have been utilized in evaluation benchmarks to judge the model responses, including both pointwise settings (Liu et al., 2023b; Yu et al., 2023a; Sun et al., 2023; Zhang et al., 2024a; Li et al., 2024a; Zhang et al., 2024b) and pairwise settings (Lu et al., 2024; Yu et al., 2024b). Our LLaVA-Critic are evaluated in the these evaluation scenarios as open-source alternative, with advantages in cheap and customized evaluation. For open-source models, Prometheus-Vision (Lee et al., 2024) is the first VLM trained as an evaluator for specific user-designed scoring criteria. While sharing the same open-source spirit, our proposed LLaVA-Critic is favored as the first open generalist evaluator. Note that GPT is also utilized to extract answers from LMM responses for subsequent evaluation in some benchmarks (Lu et al., 2023; Guan et al., 2024; Wang et al., 2024d). This extractive functionality for evaluation is out of the scope of this paper.

**Preference learning for LMMs.** Reinforcement learning from human feedback (RLHF) is a proven method to align large language models (LLMs) with human intentions. DPO (Rafailov et al., 2024) introduces a new parameterization of the reward model in RLHF, enabling direct optimization using pairwise preference datasets. CriticGPT (McAleese et al., 2024) trains "critic" models that help evaluate model-generated code, which is further utilized as feedback signals to improve code LLM. The concept of preference learning has recently expanded from language models to the multimodal space. LLaVA-RLHF (Sun et al., 2023), the first open-source work in this area, improves visual chat abilities for LMMs using human-scored rankings. Research on preference learning for LMMs has since then advanced in several studies. BPO (Pi et al., 2024) conducts preference learning by introducing negative responses generated by the model itself, using distorted images or text-based LLMs to inject errors. Wang et al. (2024a) proposes mDPO, which introduces conditional preference optimization to emphasize image information. Other works apply preference alignment to reduce hallucinations and enhance the overall capabilities of vision-language models (VLMs), either through human feedback (e.g., RLHF-V (Yu et al., 2024a)) or AI feedback (e.g., Silkie: VLFeedback (Li et al., 2023c)). Several approaches use self-rewarding mechanisms to minimize dependence on external preference pairs, such as divide-and-conquer strategies (Yu et al., 2024b) (RLAIF-V), sentence-level beam search (Zhou et al., 2024b), deliberate hallucination injection (Zhou et al., 2024a), or metric-based self-critic prompts (Wang et al., 2024c). As a generalist evaluator, LLaVA-Critic can provide valuable feedback for LMM alignment, paving the way for self-improving AI models.

## 3 DATA COLLECTION

We now introduce the data collection process for the LLaVA-Critic training dataset. The use of GPT-4/4V as a generalist evaluator for LMMs can be broadly categorized into two settings: $(i)$ **Pointwise scoring**: GPT assigns a score to an individual candidate response, either by directly evaluating it based on predefined criteria or by scoring it relative to a fixed reference answer. This setting can be regarded as a combination of the *single-answer grading* and *reference grading* methods discussed in Zheng et al. (2024). $(ii)$ **Pairwise ranking**: GPT compares two candidate responses to determine their relative quality (or declares a tie). To equip LLaVA-Critic with a generalized

**Multimodal Input for LMM:**
***Question***: What are the specifics visible in the image?

---

**Setting 1: Pointwise Scoring**
***Response***: The image shows a small train with four red cars, traveling on a track. The train is located in a park setting, and there are potted plants nearby.
***Evaluation Prompt***: From 0 to 100, how much do you rate for this Text Caption in terms of the correct and comprehensive description of the image? Do not dominant the rating by a single attribute such as recognition correctness, but a overall rating on the object/scene appearance, position, pose, action, shape, etc., and contents in the background. Do not consider the appropriateness or sensitive descriptors, such as "middle-aged western man", judge based on if it has correct specifications of the object and scenes in image. Provide a few lines for explanation and the rate number at last after "Final Score:".
**LLaVA-Critic**: The text caption provides a generally accurate description of the image. It correctly identifies the small train with four red cars and mentions the park setting and the presence of potted plants. However, there are a few details that could be improved for a more comprehensive description. For instance, the caption could mention that the train is stationary rather than traveling, and it could describe the surrounding structures, such as the building and the wooden fence, to give a fuller picture of the scene. Final Score: 85.

---

**Setting 2: Pairwise Ranking**
***Response A***: The image shows a small train on a track, with a total of six cars attached to it. The train is traveling on a small track, and there are potted plants nearby. The scene appears to be set in a park or a similar outdoor area.
***Response B***: The image shows a small train with four red cars, traveling on a track. The train is located in a park setting, and there are potted plants nearby.
***Evaluation Prompt***: "Given an image and a corresponding question, please serve as an unbiased and fair judge to evaluate the quality of the answers provided by a Large Multimodal Model (LMM). Determine which answer is better and explain your reasoning with specific details. Your task is provided as follows:
Question: []\n Response A: []\n Response B: []\n ASSISTANT:\n"
**LLaVA-Critic**: Response B is better than response A because it provides a more accurate description of the objects and their attributes in the image. Response B correctly identifies the train as having four red cars, whereas Response A inaccurately states that there are six cars. Both responses mention the potted plants and the park setting, but Response B's accuracy in the number of train cars makes it more precise. Additionally, Response B avoids unnecessary repetition and provides a clearer depiction of the scene, making it more helpful and accurate overall.

Table 1: An example of LLaVA-Critic training data. The top block shows pointwise scoring, where LLaVA-Critic predicts a score to evaluate a single response's quality; the bottom block illustrates pairwise ranking, where it rank response pairs. In both settings, LLaVA-Critic learns to provide reasons for its judgments.

evaluation capacity as with GPT-4V, we design a GPT-assisted pipeline to curate our training dataset for both settings. An example of LLaVA-Critic training data is shown in Table 1.

## 3.1 POINTWISE DATA

To train a generalist critic model for the evaluation of individual LMM responses, each pointwise training data instance consists of the following components: a multimodal instruction (i.e., a question-image pair), a model response, an optional reference answer (depending on the evaluation prompt), an evaluation prompt, a judgment score, and the corresponding justification for the score. By organizing them in a sequence, the training sample is:

```
(Image, Question, Response, Reference, Evaluation Criteria, Score, Reason),
```

where green parts are treated as model output to compute the auto-regressive loss, the order of `Score` and `Reason` is specified by the evaluation prompt. We select multimodal instructions from 8 multimodal instruction tuning datasets, spanning across a wide range of tasks including: (1) general visual conversation, detailed captioning and reasoning (LLaVA-Instruction-150k (Liu et al., 2023b), SVIT (Zhao et al., 2023)); (2) more challenging tasks such as complex reasoning (ComVint (Du et al.,

2023)), text-rich understanding (LLaVAR (Zhang et al., 2023b)) and robustness-oriented instructions (LRV-Instruction (Liu et al., 2023a)); and (3) various specific domains such as academic question answering (M3IT (Li et al., 2023d)), medical image understanding (LLaVA-Med (Li et al., 2023b)) and embodied decision-making (PCA-EVAL (Chen et al., 2023a)). For each multimodal instruction, we select one or more model responses from VLFeedback (Li et al., 2023c), which collects multiple responses from 12 off-the-shelf LMMs. Additionally, we generate responses using GPT-4o, a leading commercial LMM, to serve as high-quality reference answers.

To equip LLaVA-Critic with general evaluation capacities across various tasks, we construct an evaluation prompt pool from 7 widely used multimodal benchmarks that utilize GPT-as-a-judge, including LLaVA-in-the-Wild (Liu et al., 2023b), LLaVA-Wilder (Li et al., 2024a), Image Detailed Captioning (Li et al., 2024a), MMHal-Bench (Sun et al., 2023), MMVet (Yu et al., 2023b), WildVision-Bench (Lu et al., 2024) and RefoMB (Yu et al., 2024b). [1] Prompts that require additional textual context—since they use text-only GPT-4 as the evaluator—are adjusted to focus on the input image, better aligning with the LMM evaluator setting. To construct training data based on each evaluation prompt, we select multimodal instructions and model responses according to the specified evaluation scenario, and include reference answers from GPT-4o when necessary. These components are then assembled into the evaluation prompt and used as input for GPT-4o (as-a-judge) to provide high-quality judgment scores and detailed justifications for model responses. Finally, our pointwise training dataset comprises a total of 18,915 question-image pairs and 72,782 critic data samples.

## 3.2 PAIRWISE DATA

The pairwise data consists of responses with known preference relationships. In our training dataset, we collect the pairwise data from three open-source datasets: VLFeedback (Li et al., 2023c), RLHF (Sun et al., 2023), and RLHF-V (Yu et al., 2024a). In the VLFeedback dataset, each (question, response) pair is rated across three different dimensions by GPT-4V. For the same question, responses generated by different LMMs can form multiple response pairs for that question. We randomly select 20k pairs where the average score gap between responses is greater than 0.6. Besides, to ensure diversity in the preferences, we randomly sample 5k pairs where the two responses had identical scores across all three dimensions to serve as "Tie" training data. In the RLHF dataset, each question is annotated with preference relationships between different responses by human evaluators. In contrast, the RLHF-V dataset consists of responses generated by LMM, which have been manually refined to produce improved responses. From these two datasets, we collect 9.4k (RLHF) and 5.7k (RLHF-V) response pairs, each annotated with human preferences. This results in a total of 40.1k pairwise data samples.

To enable LLaVA-Critic to provide useful detailed feedback in addition to the preference relation, we utilize GPT-4o to generate reasons behind the given preference judgment. The training sample for pairwise data is structured in the following sequence:

(Image, Question, Response 1&2, Evaluation Criteria, Preference, Reason),

where the evaluation criteria is from carefully designed prompt templates. To allow LLaVA-Critic to handle diverse pairwise data ranking, we develop a set of 30 prompt templates (see Appendix A.1). Each preference pair is randomly assigned a template from this set, forming the final training data.

**Data statistics.** Our training dataset comprises a total of 46k images and 113k data samples. As illustrated in Figure 1, we curate our training set with diverse instruction-response pairs, spanning multiple evaluation tasks and domains.

# 4 LLAVA-CRITIC

## 4.1 MODEL

To train the LLaVA-Critic model, we fine-tune a pre-trained LMM that already possesses strong capabilities in following diverse instructions. This is crucial, as it ensures that the model has already

---

[1]Although RefoMB and WildVision-Bench use pairwise evaluation prompts, only one response is evaluated, with the other from a fixed reference model (GPT-4V and Claude-3-Sonnet, respectively), making them pointwise evaluations. In our dataset, GPT-4V responses in VLFeedback serve as reference answers for both prompts.

| Setting | Prompt source | | Data source | Data size |
|---|---|---|---|---|
| Pointwise | LLaVA-in-the-Wild | | LLaVA, SVIT, LLaVAR, LLaVAMed, ComVint | 17.5k |
| | LLaVA-Wilder | | SVIT, LLaVAR, LLaVAMed, ComVint, M3IT, PCAEval | 16.6k |
| | WildVision-Bench | | VLFeedback | 14.0k |
| | MMVet | | LLaVAR, LLaVAMed, M3IT, PCAEval | 9.3k |
| | MMHAL-Bench | | LRV-Instruction | 7.6k |
| | ImageDC | | SVIT-detail | 5.3k |
| | RefoMB | | VLFeedback | 2.5k |
| Pairwise | 30 manually crafted prompt templates | | VLFeedback | 20.0k |
| | | | LLaVA-RLHF | 9.4k |
| | | | VLFeedback (Tie) | 5.0k |
| | | | RLHF-V | 5.7k |

Figure 1: Data statistic of LLaVA-Critic-113k training dataset. In the pointwise setting, we categorize datasets by instruction sources and select data based on the task type corresponding to each evaluation prompt. Note that all our training data is sourced from public instruction-following training sets and does not overlap with with any evaluation benchmarks.

been equipped to handle a wide range of vision tasks in the wild with high quality. The evaluation ability is treated as an additional discriminative ability closely tied to these scenarios. During training, LLaVA-Critic takes an evaluation prompt—assembling the multimodal instruction input, model response(s), and an optional reference response—as input. It is trained to predict quantitative pointwise scores or pairwise rankings based on the criteria in the evaluation prompt, and provide detailed justifications for the assigned judgments. Standard cross-entropy loss is applied to both judgments and justifications.

In our experiments, we start with the LLaVA-OneVision(OV) 7B/72B pretrained checkpoint and fine-tune it on the proposed LLaVA-Critic-113k dataset for 1 epoch to develop LLaVA-Critic. We apply a learning rate of 2e-6 and a batch size of 32 for training, with other hyperparameters set to the defaults from Li et al. (2024b). We also curate a subset with 53k samples (42k pointwise, 11k pairwise) that cover fewer instruction sources and domains. The model trained on this reduced subset is referred to as LLaVA-Critic (v0.5).

## 4.2 SCENARIO 1: LMM-AS-A-JUDGE

Evaluating complex tasks often requires human judges to provide feedback, which can be labor-intensive. LLaVA-Critic can serve as a general evaluator for LMM responses, reducing labor costs by automating the evaluation process. LLaVA-Critic consistently provides reliable judgments and justifications aligned with GPT-4o or human evaluations across a range of widely used multimodal benchmarks. This consistency holds true for both instance-level scoring and model-level ranking, as demonstrated in Sec. 5.1.

Specifically, we consider the following evaluation scenarios: ($i$) *Visual Chat*. This task involves handling daily-life visual tasks through multimodal dialogue, requiring evaluation of task completion quality in a conversation setting. Examples include LLaVA-Bench (Liu et al., 2023b) and LLaVA-in-the-Wild (Liu et al., 2023b), which focus on simpler scenarios, while LLaVA-Wilder (Li et al., 2024a) addresses more challenging cases. ($ii$) *Integrated capabilities*. Real-world tasks require integration of multiple basic abilities of LMMs. MM-Vet (Yu et al., 2023b) offers a comprehensive benchmark, evaluating core vision-language capabilities including recognition, OCR, knowledge integration, language generation, spatial awareness, and math. The Multimodal Live-Bench tests the model's ability to generalize to new, unobserved knowledge by leveraging continuously updated news and online forums. ($iii$) *Preferences*. This task simulates real-world multimodal interactions where models are expected to align their behavior with human preferences. The WildVision-Bench (Lu et al., 2024) is a prime example, replicating scenarios from the online platform WildVision-Arena (WV-Arena) to evaluate preference-based interactions. ($iv$) *Detailed Description*. This task assesses models on their ability to provide comprehensive and detailed descriptions of images and videos. Image Detailed Captioning (Li et al., 2024a) evaluates detailed descriptions in images, while video Detailed Captioning (Zhang et al., 2024c) extends these capabilities from images to video contexts. ($v$) *Hallucination*: This task focuses on the model's ability to provide grounded responses based on the given context, ensuring that it avoids generating inaccurate or fabricated information, exemplified by MMHal-Bench (Sun et al., 2023).

### 4.3 SCENARIO 2: PREFERENCE LEARNING

Leveraging a generalist evaluator as a critic to generate reward signals for reinforcement learning is a promising research direction. In this work, we employ LLaVA-Critic to produce AI-generated feedback datasets for diverse tasks, thereby improving the performance of supervised fine-tuned LMMs through preference alignment. Notably, the reward signals generated by our critic can be utilized in any preference learning algorithms, including RLHF and DPO. To quickly assess the effectiveness of the reward data, we focus on how LLaVA-Critic is incorporated into the iterative DPO training process.

- **Step 1: Response generation**. The iterative DPO process begins with a pretrained LMM $\pi_0$ as the initial checkpoint and a set of multimodal instructions $\{(\boldsymbol{x}_k, \boldsymbol{v}_k)\}_{k=1}^{N}$, where each $\boldsymbol{x}_k$ is a question and $\boldsymbol{v}_k$ is the corresponding image. For each question-image pair $(\boldsymbol{x}, \boldsymbol{v})$, the pretrained LMM $\pi_0$ randomly generates $K$ responses $\{y_1, y_2, \ldots, y_K\}$, sampled independently from its distribution.
- **Step 2: Scoring**. To mitigate order-related variance in LLaVA-Critic's preferences, we form all possible ordered pairs from these responses, resulting in $K \times (K-1)$ pairs. For each response pair $(y_i, y_j)$, we apply LLaVA-Critic with an evaluation prompt to generate a relative score $a_{ij}$, which normalizes the score of $y_j$ based on $y_i$.
- **Step 3: Reward Preference**. The overall reward score $r_i$ for each response $y_i$ is calculated by aggregating these preference scores: $r_i = \sum_{k \neq i} a_{ki} - \sum_{l \neq i} a_{il}$ This calculation effectively measures how much better or worse $y_i$ is compared to all other responses. We then select the responses with the highest and lowest reward scores as the best and worst responses, denoted as $y^+$ and $y^-$, respectively. These form the pairwise feedback data $(y^+, y^-)$, which is used for DPO training to enhance the LMM's alignment with LLaVA-Critic's preferences.

**Iterative Improvement.** After each round of DPO training, the updated LMM becomes the new starting checkpoint. The process is then repeated iteratively for another $M-1$ rounds, using LLaVA-Critic to progressively improve the model's performance based on its self-generated responses.

## 5 EXPERIMENTAL RESULTS

### 5.1 LMM-AS-A-JUDGE

To comprehensively assess the LLaVA-Critic's capacity in evaluating LMM responses across different scenarios, we consider two primary experimental settings: (1) *In-domain Judgments*: where we measure LLaVA-Critic's consistency with GPT-4o or human evaluators on evaluation tasks/prompts included in the LLaVA-Critic-113k training dataset; and (2) *Out-of-domain Judgments*: where we apply LLaVA-Critic on evaluation tasks and prompts that are unseen during training. For the second setting, we use the MLLM-as-a-Judge (Chen et al., 2024) benchmark to assess the alignment between LLaVA-Critic and human evaluators in generalized scenarios.

**In-domain Pointwise Scoring** To evaluate the consistency between LLaVA-Critic and GPT-4o (OpenAI, 2024b) in pointwise scoring across different evaluation scenarios, as described in Sec. 4.2, we select 7 popular multimodal benchmarks and collect candidate responses from 13 commonly used LMMs alongside their GPT-4o evaluations, resulting in a total of 14174 examples (see details in Appendix A.2). LLaVA-Critic is then tasked with providing judgments on theses samples. We report Pearson correlation and Kendall's Tau to measure the degree of alignment with GPT-4o in terms of instance-wise scoring and model-wise ranking respectively.

We conduct experiments based on three different baseline models: LLaVA-NeXT (LLaMA-8B) (Liu et al., 2024b; Li et al., 2024a), LLaVA-OneVision-7B, and LLaVA-OneVision-72B. The experimental results are shown in Table 2. Across all models and benchmarks, LLaVA-Critic variants significantly improve their corresponding baseline models in both Pearson-r and Kendall's Tau. ($i$) *Data scaling*. By comparing the performance between v0.5 and full data trained LLaVA-Critic-7B, it concludes the necessity of larger size and diversity of instruction in training data. ($ii$) *Model scaling*. The best performance in terms of Pearson-r is achieved by LLaVA-Critic-72B with an average score of 0.754, which significantly outperforms the LLaVA-OV-72B baseline (0.634). Similarly, in Kendall's Tau, LLaVA-Critic-72B achieves the highest average score of 0.933, again outperforming the LLaVA-OV-72B baseline (0.802). This indicates that LLaVA-Critic-72B already possesses pointwise scoring

| LMM Evaluator | Pearson-r (↑) | | | | | | | |
|---|---|---|---|---|---|---|---|---|
| | ImageDC | MMVet | WildVision | LLaVA-B | LLaVA-W | L-Wilder | MMHal | Avg. |
| LLaVA-NeXT (LLaMA-8B) | 0.262 | 0.317 | 0.147 | 0.211 | 0.345 | 0.156 | 0.472 | 0.273 |
| LLaVA-Critic (LLaVA-NeXT) | 0.673 | 0.706 | 0.580 | 0.529 | 0.820 | 0.936 | 0.748 | 0.713 |
| LLaVA-OV-7B | 0.056 | 0.349 | 0.251 | 0.335 | 0.533 | 0.592 | 0.433 | 0.364 |
| LLaVA-Critic-7B (v0.5) | 0.737 | 0.718 | 0.571 | 0.494 | 0.789 | 0.932 | 0.746 | 0.712 |
| LLaVA-Critic-7B | 0.735 | 0.733 | 0.616 | 0.510 | 0.843 | 0.940 | 0.748 | 0.732 |
| LLaVA-OV-72B | 0.718 | 0.680 | 0.446 | 0.436 | 0.716 | 0.824 | 0.620 | 0.634 |
| LLaVA-Critic-72B | 0.802 | 0.723 | 0.705 | 0.524 | 0.782 | 0.951 | 0.790 | 0.754 |

| LMM Evaluator | Kendall's Tau (↑) | | | | | | | |
|---|---|---|---|---|---|---|---|---|
| | ImageDC | MMVet | WildVision | LLaVA-B | LLaVA-W | L-Wilder | MMHal | Avg. |
| LLaVA-NeXT (LLaMA-8B) | 0.452 | 0.436 | 0.615 | 0.487 | 0.503 | 0.231 | 0.590 | 0.473 |
| LLaVA-Critic (LLaVA-NEXT) | 0.787 | 0.974 | 0.846 | 0.839 | 0.923 | 0.974 | 0.923 | 0.895 |
| LLaVA-OV-7B | 0.539 | 0.154 | 0.795 | 0.667 | 0.641 | 0.839 | 0.590 | 0.603 |
| LLaVA-Critic-7B (v0.5) | 0.813 | 0.897 | 0.872 | 0.846 | 0.949 | 0.974 | 0.923 | 0.896 |
| LLaVA-Critic-7B | 0.897 | 0.949 | 0.897 | 0.839 | 0.923 | 0.974 | 0.897 | 0.911 |
| LLaVA-OV-72B | 0.872 | 0.795 | 0.821 | 0.667 | 0.769 | 0.949 | 0.744 | 0.802 |
| LLaVA-Critic-72B | 0.949 | 0.949 | 0.949 | 0.821 | 0.923 | 0.994 | 0.949 | 0.933 |

Table 2: Comparisons on in-domain pointwise scoring. LLaVA-Critic consistently outperforms other baseline methods across 7 multimodal evaluation benchmarks.

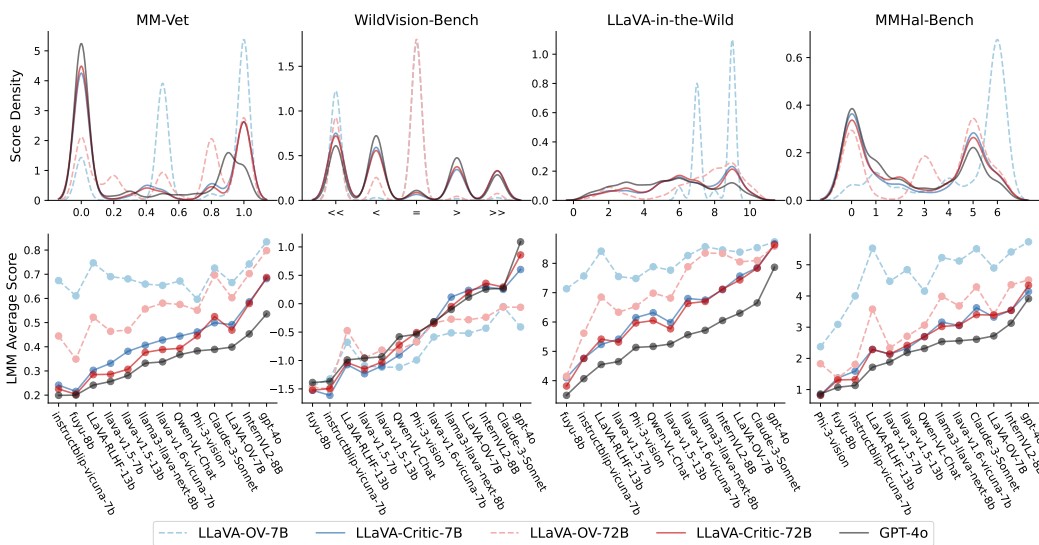

Figure 2: *(Top):* Overall distribution of evaluation scores across 4 benchmarks. *(Bottom):* Calculated average evaluation score for each response model on each benchmark. Each color represents a different LMM evaluator. Leveraging high-quality critic training data, LLaVA-Critic closely aligns with GPT-4o in delivering balanced evaluation scores and accurately ranking response LMMs.

capabilities that are quite aligned with GPT-4o. Moreover, it is worth noting that even with a significant reduction in model parameters, LLaVA-Critic-7B still exhibits very strong point-wise scoring capabilities. With a Pearson-r of 0.732 and a Kendall's Tau of 0.911, its performance has not decreased significantly compared to LLaVA-Critic-72B. This presents an advantage for deploying and utilizing LLaVA-Critic in resource-constrained environments.

Figure 2 provides a qualitative comparison between LLaVA-Critic and other LMM evaluators. While LLaVA-OneVision often assigns fixed scores (e.g., "Tie" on WildVision-Bench or "6" on MMHal-Bench), LLaVA-Critic produces more diverse and balanced scores that closely align with GPT-4o, leading to consistent rankings of response models. Notably, even without training on critic data, LLaVA-OneVision-72B demonstrates model-wise rankings that partially align with GPT-4o across four multimodal benchmarks.

| Model | Accuracy w. Tie↑ | Accuracy w.o. Tie↑ | Kendall's Tau↑ |
|---|---|---|---|
| GPT-4o | 0.617 | 0.734 | 0.819 |
| GPT-4V | 0.620 | 0.733 | 0.787 |
| LLaVA-NeXT (LLaMA-8B) | 0.473 | 0.569 | 0.605 |
| LLaVA-OV-7B | 0.531 | 0.640 | 0.715 |
| LLaVA-OV-72B | 0.594 | 0.708 | 0.763 |
| LLaVA-Critic-7B (v0.5) | 0.580 | 0.692 | 0.755 |
| LLaVA-Critic(LLaVA-NeXT) | 0.582 | 0.686 | 0.755 |
| LLaVA-Critic-7B | 0.596 | 0.722 | 0.763 |
| LLaVA-Critic-72B | 0.605 | 0.736 | 0.779 |

Table 3: Comparisons on in-domain pairwise ranking. LLaVA-Critic is comparable with GPT-4V in terms of alignment with human evaluators.

**In-domain Pairwise Ranking**  To assess the consistency between LLaVA-Critic and human evaluators in pairwise ranking, we use the battle data from WildVision Arena (Lu et al., 2024), which comprises 11k human-annotated preference relations among LMM response pairs. Each relation includes a question-image pair and two responses generated by different models, accompanied by a human-annotated preference (including ties). From this dataset, we randomly sample 2k response pairs and assign them to evaluation prompts from the pairwise ranking prompt template set mentioned in Section 3.2, creating the in-domain evaluation dataset. We report average accuracy, with and without ties, to assess alignment with human evaluators at the instance level. For model-level consistency, we calculate the Elo rating for each response LMM and report Kendall's Tau to measure the overall ranking correlation with human preferences.

Experimental results are reported in Table 3. While the LLaVA models exhibit initial pairwise ranking ability, there is a notable performance gap compared to GPT-4V/4o. After training with critic data, LLaVA-Critic achieves significant improvements. Specifically, LLaVA-Critic-72B achieves an average accuracy of 73.6% in pairwise comparisons without tie, outperforming both GPT-4o and GPT-4V. For pairwise comparison with tie (Accuracy w. Tie) and model-wise ranking (Kendall's Tau), LLaVA-Critic-72B shows only a marginal gap compared to GPT-4V/4o, with an accuracy of 60.5% and a score of 0.779, respectively. Notably, despite a substantial reduction in the number of parameters, LLaVA-Critic-7B still achieves an average accuracy of 59.6% in pairwise ranking with ties and 72.2% without ties, alongside a Kendall's tau of 0.763. These results underscore the strong alignment between LLaVA-Critic and human evaluators in pairwise ranking LMM responses.

**MLLM-as-a-Judge**  MLLM-as-a-Judge (Chen et al., 2024) is a comprehensive benchmark to evaluate the degree of alignment between model-based evaluation and human evaluation. It collects approximately 17k image-instruction-response triplets across 14 multimodal benchmarks and 6 LMM response models. Human annotators are then employed to assess model responses under scoring, pairwise comparison and batch ranking settings, resulting in 7756, 5719, 1469 examples respectively. In our experiments, we evaluate LLaVA-Critic in both (pointwise) scoring and pair comparison settings to assess its general alignment with human evaluators. We report the average Pearson correlation for scoring and average accuracy for pairwise comparison, following the metrics used in the original benchmark.

We compare LLaVA-Critic with commercial models (GPT-4V/4o, Gemini-Pro (Team et al., 2023)), open-sourced LMMs, as well as Prometheus-Vision (Lee et al., 2024), which trains a LLaVA model on a curated LMM-as-a-judge dataset comprising 15k GPT-generated rubrics and 150k GPT-4V feedback data. As shown in Table 4, LLaVA-Critic-7B surpasses all baselines except GPT-4V/4o across all settings by a considerable margin. Built on a stronger base model, LLaVA-Critic-72B further achieves the Pearson similarity with human annotators from 0.314 to 0.393 in pointwise scoring. For pairwise comparisons, it achieves accuracy rates of 57.8% and 71.5% with and without ties, respectively, reaching a level of alignment with human evaluators comparable to GPT-4V/4o. We also compare different variants of LLaVA-Critic and observe performance gains with both stronger base models and larger training data, consistent with previous findings. This again highlights the critical role of model and data scaling in building an effective and generalist open-source LMM evaluator. More comprehensive results are provided in Appendix B.1.

**Qualitative Comparison.**  We present example comparisons of the evaluation scores and reasons generated by LLaVA-Critic and other LMMs, with detailed examples provided in Appendix C. The

| Model | Score↑ | Pair w. Tie↑ | Pair w.o. Tie↑ |
|---|---|---|---|
| GPT-4V* | 0.490 | 0.636 | 0.773 |
| GPT-4o[†] | 0.439 | 0.577 | 0.736 |
| GPT-4V[†] | 0.424 | 0.538 | 0.717 |
| Gemini-pro* | 0.304 | 0.509 | 0.615 |
| LLaVA-v1.5-7B | 0.158 | 0.439 | 0.576 |
| LLaVA-NeXT (LLaMA-8B) | 0.198 | 0.461 | 0.586 |
| LlaVA-OV-7B | 0.151 | 0.426 | 0.550 |
| LlaVA-OV-72B | 0.287 | 0.513 | 0.701 |
| Prometheus-Vision (LLaVA-v1.5-7B) | 0.213 | – | – |
| LLaVA-Critic (LLaVA-v1.5-7B) | 0.228 | 0.528 | 0.656 |
| LLaVA-Critic (LLaVA-NeXT) | 0.272 | 0.547 | 0.677 |
| LLaVA-Critic-7B (v0.5) | 0.312 | 0.546 | 0.675 |
| LLaVA-Critic-7B | 0.314 | 0.556 | 0.689 |
| LLaVA-Critic-72B | 0.393 | 0.578 | 0.715 |

Table 4: Results on MLLM-as-a-Judge (Chen et al., 2024). *: the results as reported in the original paper (Chen et al., 2024); [†]: results from our evaluation of GPT-4V/4o based on their codebase. Note that Prometheus-Vision cannot follow the pairwise evaluation prompt. LLaVA-Critic significantly narrows the gap between open-source LMMs and GPT-4V/4o in their ability to evaluate LMM responses across a range of evaluation scenarios.

key findings are as follows: Compared to LLaVA-OneVision, LLaVA-Critic delivers more accurate judgments (Table 10), and provides more concrete, image-grounded justifications (Table 11). The latter is crucial for reliable AI (Bai et al., 2022), as offering well-supported reasons for evaluations establishes LLaVA-Critic as a transparent evaluator of LMM responses.

## 5.2 PREFERENCE LEARNING

We further evaluate LLaVA-Critic's performance in providing reward signals for iterative DPO. LLaVA-OneVision's supervised fine-tuned checkpoint is used as the base policy model, and question-image pairs from LLaVA-RLHF (Sun et al., 2023) serve as the multimodal instructions. For each pair, $K = 5$ candidate responses are generated through random decoding (with a temperature of 0.7 and top-p of 0.9) to ensure response diversity. LLaVA-Critic is employed as described in Sec. 4.3 to construct the pairwise feedback data, which is then used for one epoch of DPO training. We perform iterative DPO for $M = 3$ rounds in total.

To assess the effectiveness of the LLaVA-Critic's reward signals, we evaluate the final LMM checkpoint across 6 open-ended multimodal benchmarks: four image-based tasks (LLaVA-in-the-Wild (Liu et al., 2023b), LLaVA-Wilder (Li et al., 2024a), LiveBench (Zhang et al., 2024a), and WildVision-Bench (Lu et al., 2024)), one video-based task (Video Detailed Captioning (Li et al., 2024a)), and one hallucination benchmark (MMHal-Bench (Sun et al., 2023)). We compare LLaVA-Critic with two baselines: (1) the reward model from LLaVA-RLHF (Sun et al., 2023), which is trained on human preferences, and (2) a naive baseline that replaces LLaVA-Critic with LLaVA-OneVision's SFT checkpoint as a zero-shot reward model.

As shown in Table 5, preferences provided by LLaVA-Critic significantly improve LLaVA-OneVision's visual chat capacities and reduce hallucination across challenging tasks. LLaVA-Critic consistently surpasses other baseline reward models on 5 out of 6 benchmarks for the 7B base model and all 6 benchmarks for the 72B base model. Despite the preference alignment conducted solely with images, LLaVA-Critic also enhances LLaVA-OneVision's performance in Video Detailed Captioning (+0.12 on OV-7B and +0.26 on OV-7B), demonstrating its ability to generalize to both image and video contexts. Additionally, we observe that Critic-7B outperforms Critic-7B-v0.5 on 5 out of 6 benchmarks, highlighting the importance of stronger reward models—trained on more diverse critic instructions—to deliver more accurate reward signals and further enhance preference learning. Please refer to Appendix B.2 for additional results and Table 12 for a visual-chat example.

**Comparison** We take LLaVA-v.1.5-7B as the base policy model, and compare LLaVA-Critic with 4 previous methods that apply preference optimization with self-generated candidate responses: LLaVA-

| Base | Reward Signal | LLaVA-W↑ | L-Wilder↑ | WildVision↑ | LiveBench↑ | VideoDC↑ | MMHal↑ |
|---|---|---|---|---|---|---|---|
| GPT-4V | – | 98.0 | 81.0 | 79.8 | 73.7 | 4.00 | 3.83 |
| OV-7B | – | 90.7 | 67.8 | 54.0 | 77.1 | 3.75 | 3.19 |
| | OV-7B | 98.6 | 70.9 | 66.6 | 84.0 | 3.77 | 3.79 |
| | LLaVA-RLHF | 97.5 | 70.3 | 64.1 | 83.1 | 3.84 | **4.01** |
| | Critic-7B (v0.5) | 98.1 | 70.5 | 67.2 | **85.1** | 3.83 | 3.85 |
| | Critic-7B | **100.3** | **71.6** | **67.3** | 84.5 | **3.87** | 3.91 |
| OV-72B | – | 93.5 | 72.0 | 51.7 | 81.5 | 3.60 | 3.61 |
| | LLaVA-RLHF | 103.2 | 75.2 | 65.2 | 86.2 | 3.85 | 3.67 |
| | Critic-72B | **104.4** | **75.9** | **70.0** | **88.5** | **3.86** | **3.77** |

Table 5: Comparison between LLaVA-Critic and other baselines in preference alignment. "Base" refers to the initial LMM checkpoint for DPO. For both LLaVA-OV-7B and LLaVA-OV-72B base models, iterative DPO training with LLaVA-Critic's reward signal leads to more significant performance gains across various multimodal benchmarks.

| Method | #Prompts | LLaVA-W | L-Wilder | WildVision | LiveBench | MMHal* | $MME^P$ | $MME^C$ | MMB-en | MM-Vet | MMStar |
|---|---|---|---|---|---|---|---|---|---|---|---|
| LLaVA-v1.5-7B | – | 63.4 | 54.2 | 20.4 | 45.6 | 1.94 | 1510.7 | 348.2 | 64.3 | 30.5 | 33.3 |
| + RLHF | 9.4k | 63.7 | 54.5 | 19.8 | 46.2 | 1,90 | 1508.2 | 360.2 | 60.4 | 31.1 | 33.0 |
| + SIMA | 17k | 66.1 | 52.3 | 17.6 | 47.9 | 1.81 | 1507.7 | **379.3** | 64.9 | 31.6 | 34.7 |
| + CSR | 15k | 71.1 | 55.9 | 20.0 | 45.0 | 1.96 | **1524.2** | 367.9 | **65.4** | 33.9 | 33.6 |
| + RLAIF-V | 33.8k | 72.7 | 56.4 | 19.2 | **50.4** | **3.04** | 1362.7 | 302.9 | 62.6 | 26.7 | **35.4** |
| + LLaVA-Critic | 9.4k | **73.5** | **57.2** | **29.2** | 50.0 | 2.07 | 1500.4 | 350.7 | 64.1 | 32.2 | 34.2 |

Table 6: Comparison with other preference learning algorithms on LLaVA-v1.5-7B. Apart from benchmarks in Table 5, we also report the results on 5 comprehensive multimodal benchmarks for reference. The best and second best results are shown in **bold** and underlined respectively. *OpenAI's *gpt-4-0613* is used for the MMhal-Bench evaluation due to the deprecation of the original API.

RLHF (Sun et al., 2023), SIMA (Wang et al., 2024c), CSR (Zhou et al., 2024b) and RLAIF-V (Yu et al., 2024b). These methods primarily vary in the source of reward signals: LLaVA-RLHF leverages a pretrained reward model based on human feedback; SIMA develops an in-context self-critic prompt for providing pairwise judgments; CSR incorporates sentence-level beam search with CLIP-score calibration; and RLAIF-V adopts a divide-and-conquer strategy to calculate the overall reward score by combining sentence-level judgments. For our method, we utilize the prompts (question-image pairs) from the LLaVA-RLHF dataset and perform DPO training for 3 epochs.

As illustrated in Table 6, with only 9.4k input prompts, the reward signal provided by LLaVA-Critic substantially improve the base model's performance across various open-ended visual chat benchmarks. It achieves the best improvements of +10.1 on LLaVA-W, +3.0 on LLaVA-Wilder, +8.8 on WildVision, along with the second-highest gains of + 4.4 on LiveBench and +0.13 on MMHal-Bench, respectively. At the same time, the overall capacities of LLaVA-v1.5-7B are largely preserved, as demonstrated on other comprehensive benchmarks. This is superior to other competing methods, which either result in smaller performance gains or achieve improvements by compromising the overall capabilities on other benchmarks.

## 6 CONCLUSIONS

We have presented LLaVA-Critic, an open-source LMM that is trained to evaluate model performance in a wide range of multimodal scenarios. To achieve this, we curated a high-quality critic instruction-following dataset with diverse evaluation criteria. We demonstrated the effectiveness of LLaVA-Critic in two key areas: (1) as a generalized evaluator, LLaVA-Critic provides pointwise scores and pairwise rankings that closely align with human and GPT-4o preferences across multiple evaluation tasks, presenting a viable open-source alternative to commercial GPT models for autonomous assessment of open-ended LMM responses; (2) in preference learning, LLaVA-Critic functions as a reliable reward model, supplying preference signals that enhance the visual chat capabilities of LMMs, surpassing the LLaVA-RLHF reward model built with human feedback. This work represents an important step toward harnessing the self-critique capabilities of open-source LMMs, and we hope it will encourage further research into developing strong LMMs with scalable and superhuman alignment feedback.

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

## A IMPLEMENTATION DETAILS

### A.1 EVALUATION PROMPTS FOR LLaVA-CRITIC TRAINING

**Pointwise prompts**  To construct pointwise training data, we adapt the existing evaluation prompts in 7 widely used multimodal evaluation benchmarks that employ GPT-as-a-judge. For further details, please refer to their papers or codebases as listed below:

- *LLaVA-in-the-Wild* (Liu et al., 2023b): arxiv.org/abs/2304.08485
- *LLaVA-Wilder* (Li et al., 2024a): llava-vl.github.io/blog/2024-05-10-llava-next-stronger-llms/
- *ImageDC* (Li et al., 2024a): llava-vl.github.io/blog/2024-05-10-llava-next-stronger-llms/
- *MMHal-Bench* (Sun et al., 2023): arxiv.org/abs/2309.14525
- *MM-Vet* (Yu et al., 2023b): arxiv.org/abs/2308.02490
- *WildVision-Arena* (Lu et al., 2024): arxiv.org/abs/2406.11069
- *RefoMB* (Yu et al., 2024b): harxiv.org/abs/2405.17220

LLaVA-in-the-Wild, MM-Vet, MMHal-Bench and RefoMB use text-only GPT models for evaluation. We slightly adjust their evaluation prompts to focus on visual information from the input image, rather than text-based contexts.

**Pairwise prompt pool**  To fully develop LLaVA-Critic's capacity of ranking LMM responses pairs across diverse scenarios, with varying format requirements and evaluation criteria, we design a set of 30 pairwise prompt templates for constructing our pairwise training data. Due to page limits, two representative prompts are provided in Table 7.

### A.2 BENCHMARK CONSTRUCTION FOR LMM-AS-A-JUDGE

**In-domain pointwise scoring**  To evaluate the ability of LLaVA-Critic in judging LMM-generated responses across varying performance levels, we select 13 off-the-shelf LMMs spanning across a wide range of visual chat capabilities, then collect their responses on 7 multimodal benchmarks. The selected response models are listed below:

- GPT-4o (OpenAI, 2024b), Claude3-Sonnet (Anthropic, 2024), LLaVA-NeXT (LLaMA-8B) (Liu et al., 2024b), LLaVA-NeXT (Vicuna-7B) (Liu et al., 2024b), LLaVA-OneVision-7B (Li et al., 2024b), LLaVA-RLHF-13B (Sun et al., 2023), LLaVA-v1.5-7B (Liu et al., 2024a), LLaVA-v1.5-13B (Liu et al., 2024a), InstructBLIP-Vicuna-7B (Dai et al., 2024), InternVL2-8B (Chen et al., 2023b), Phi-3-Vision-128k-Instruct (Abdin et al., 2024), fuyu-8B (Bavishi et al., 2023) and Qwen-VL-Chat (Bai et al., 2023)

## B ADDITIONAL EXPERIMENTAL RESULTS

### B.1 RESULTS ON MLLM-AS-A-JUDGE

We include the comprehensive results on MLLM-as-a-Judge (Chen et al., 2024) in Table 8. Since the complete evaluation pipeline has not been released, we re-implement the evaluation code based on their official codebase.

### B.2 RESULTS ON PREFERENCE LEARNING

We now present the comprehensive results of applying multi-round iterative DPO, with LLAVA-Critic providing the reward signals. As shown in Table 9, this approach significantly enhances LLaVA-OneVision's performance across six open-ended visual chat tasks, with consistent gains observed after each training round. For other comprehensive benchmarks, the iterations offer minimal benefit, while LLaVA-OneVision's overall capabilities remain largely preserved. A visualization of the performance gains on the visual chat benchmarks is provided in Figure 3.

---

**Prompt Template 1:**

You will be given an image and a corresponding question. Your task is to evaluate the responses provided, select the better response, and give reasons for your choice. Consider the following factors when making your decision.

1. *Accuracy in Object Description*: Evaluate the accuracy of the descriptions concerning the objects mentioned in the ground truth answer. Responses should minimize the mention of objects not present in the ground truth answer, and inaccuracies in the description of existing objects.

2. *Accuracy in Depicting Relationships*: Consider how accurately the relationships between objects are described compared to the ground truth answer. Rank higher the responses that least misrepresent these relationships.

3. *Accuracy in Describing Attributes*: Assess the accuracy in the depiction of objects' attributes compared to the ground truth answer. Responses should avoid inaccuracies in describing the characteristics of the objects present.

4. *Helpfulness*: Consider whether the generated text provides valuable insights, additional context, or relevant information that contributes positively to the user's comprehension of the image. Assess whether the language model accurately follows any specific instructions or guidelines provided in the prompt. Evaluate the overall contribution of the response to the user experience.

5. *Ethical Considerations*: Identify if the model gives appropriate warnings or avoids providing advice on sensitive topics, such as medical images. Ensure the model refrains from stating identification information in the image that could compromise personal privacy. Evaluate the language model's responses for fairness in treating individuals and communities, avoiding biases. Assess for harmfulness, ensuring the avoidance of content that may potentially incite violence, be classified as NSFW (Not Safe For Work), or involve other unmentioned ethical considerations. Consider any content that could be deemed offensive, inappropriate, or ethically problematic beyond the explicitly listed criteria.

The question and responses are given as follows:
Question: [] \n Response A: [] \n Response B: [] \n ASSISTANT:

---

**Prompt Template 2:**

As an expert, you are asked to evaluate two responses to the given image-based question. Provide a professional assessment of responses and decide which one is better. Support your decision with detailed reasons. Here are the question and responses:
Question: [] \n The first response: [] \n The second response: [] \n ASSISTANT:

Table 7: Example prompt templates for constructing pairwise training data. The first template covers scenarios with specific *user-defined criteria*, while the second is for general evaluations without additional instructions.

| Settings | MLLM | COCO | C.C. | Diff. | Graphics | Math | Text | WIT | Chart | VisIT | CC-3M | M2W | SciQA | Aes | MM-Vet | Ave. |
|---|---|---|---|---|---|---|---|---|---|---|---|---|---|---|---|---|
| | GPT-4V* | 0.454 | 0.507 | 0.458 | 0.645 | 0.606 | 0.624 | 0.579 | 0.645 | 0.620 | 0.431 | 0.185 | 0.383 | 0.401 | 0.326 | 0.490 |
| | Gemini-Pro* | 0.262 | 0.408 | - | 0.400 | 0.228 | 0.222 | 0.418 | 0.343 | 0.336 | 0.374 | 0.324 | 0.073 | 0.360 | 0.207 | 0.304 |
| | LLaVA-1.5-13b* | 0.247 | 0.227 | 0.060 | 0.242 | 0.093 | 0.245 | 0.109 | 0.237 | 0.177 | 0.071 | 0.424 | 0.279 | 0.414 | 0.322 | 0.225 |
| | GPT-4o† | 0.396 | 0.452 | 0.341 | 0.464 | 0.460 | 0.564 | 0.408 | 0.573 | 0.589 | 0.305 | 0.262 | 0.569 | 0.421 | 0.342 | 0.439 |
| | GPT-4V† | 0.410 | 0.444 | 0.361 | 0.449 | 0.486 | 0.506 | 0.457 | 0.585 | 0.554 | 0.266 | 0.267 | 0.315 | 0.472 | 0.367 | 0.424 |
| Score (↑) | LLaVA-v1.5-7B | 0.205 | 0.084 | 0.094 | -0.028 | 0.099 | 0.235 | -0.076 | 0.212 | 0.300 | 0.167 | 0.392 | 0.033 | 0.354 | 0.147 | 0.158 |
| | LLaVA-NeXT (LLaMA-8B) | 0.288 | 0.279 | 0.066 | 0.249 | 0.077 | 0.145 | -0.051 | 0.197 | 0.304 | 0.198 | 0.364 | 0.128 | 0.362 | 0.165 | 0.198 |
| | LLaVA-OV-7B | 0.224 | 0.024 | 0.063 | 0.189 | 0.097 | 0.265 | -0.135 | 0.274 | 0.227 | 0.081 | 0.030 | 0.261 | 0.249 | 0.262 | 0.151 |
| | LLaVA-OV-72B | 0.264 | 0.390 | 0.046 | 0.262 | 0.358 | 0.327 | 0.195 | 0.290 | 0.415 | 0.144 | 0.359 | 0.267 | 0.444 | 0.253 | 0.287 |
| | Prometheus-Vision (LLaVA-v1.5-7B) | 0.289 | 0.342 | 0.106 | 0.172 | 0.182 | 0.214 | 0.209 | 0.224 | 0.226 | 0.228 | 0.089 | 0.174 | 0.368 | 0.157 | 0.213 |
| | LLaVA-Critic (LLaVA-v1.5-7B) | 0.283 | 0.295 | 0.095 | 0.225 | 0.246 | 0.257 | 0.191 | 0.194 | 0.253 | 0.241 | 0.198 | 0.188 | 0.327 | 0.198 | 0.228 |
| | LLaVA-Critic (LLaVA-NeXT) | 0.272 | 0.406 | 0.118 | 0.257 | 0.309 | 0.275 | 0.292 | 0.354 | 0.374 | 0.225 | 0.224 | 0.091 | 0.432 | 0.176 | 0.272 |
| | LLaVA-Critic-7B (v0.5) | 0.369 | 0.456 | 0.108 | 0.291 | 0.325 | 0.340 | 0.222 | 0.388 | 0.303 | 0.205 | 0.232 | 0.270 | 0.511 | 0.338 | 0.312 |
| | LLaVA-Critic-7B | 0.382 | 0.450 | 0.103 | 0.316 | 0.356 | 0.378 | 0.179 | 0.421 | 0.322 | 0.246 | 0.301 | 0.269 | 0.395 | 0.272 | 0.314 |
| | LLaVA-Critic-72B | 0.333 | 0.463 | 0.146 | 0.452 | 0.474 | 0.559 | 0.396 | 0.545 | 0.488 | 0.273 | 0.259 | 0.334 | 0.403 | 0.374 | 0.393 |
| | GPT-4V* | 0.696 | 0.824 | 0.847 | 0.639 | 0.564 | 0.673 | 0.679 | 0.657 | 0.640 | 0.612 | 0.521 | 0.415 | 0.606 | 0.529 | 0.636 |
| | Gemini-Pro* | 0.616 | 0.787 | - | 0.650 | 0.436 | 0.664 | 0.605 | 0.500 | 0.660 | 0.560 | 0.370 | 0.262 | 0.190 | 0.312 | 0.509 |
| | Qwen-vl-plus* | 0.479 | 0.507 | 0.650 | 0.450 | 0.328 | 0.522 | 0.500 | 0.380 | 0.453 | 0.383 | 0.577 | 0.321 | 0.601 | 0.457 | 0.472 |
| | GPT-4o† | 0.582 | 0.665 | 0.829 | 0.625 | 0.433 | 0.477 | 0.565 | 0.355 | 0.577 | 0.586 | 0.581 | 0.427 | 0.873 | 0.505 | 0.577 |
| | GPT-4V† | 0.539 | 0.634 | 0.668 | 0.632 | 0.459 | 0.495 | 0.536 | 0.369 | 0.591 | 0.544 | 0.544 | 0.389 | 0.620 | 0.517 | 0.538 |
| Pair w. Tie (↑) | LLaVA-v1.5-7B | 0.460 | 0.506 | 0.577 | 0.452 | 0.302 | 0.454 | 0.375 | 0.383 | 0.518 | 0.478 | 0.495 | 0.281 | 0.466 | 0.402 | 0.439 |
| | LLaVA-NeXT (LLaMA-8B) | 0.422 | 0.595 | 0.435 | 0.404 | 0.343 | 0.431 | 0.428 | 0.384 | 0.508 | 0.496 | 0.571 | 0.336 | 0.588 | 0.512 | 0.461 |
| | LLaVA-OV-7B | 0.334 | 0.471 | 0.539 | 0.397 | 0.318 | 0.398 | 0.324 | 0.374 | 0.444 | 0.438 | 0.556 | 0.334 | 0.577 | 0.456 | 0.426 |
| | LLaVA-OV-72B | 0.464 | 0.593 | 0.667 | 0.531 | 0.434 | 0.485 | 0.447 | 0.394 | 0.549 | 0.497 | 0.557 | 0.428 | 0.596 | 0.541 | 0.513 |
| | LLaVA-Critic (LLaVA-v1.5-7B) | 0.564 | 0.674 | 0.633 | 0.505 | 0.422 | 0.528 | 0.538 | 0.386 | 0.583 | 0.608 | 0.577 | 0.294 | 0.681 | 0.404 | 0.528 |
| | LLaVA-Critic (LLaVA-NeXT) | 0.583 | 0.684 | 0.704 | 0.562 | 0.438 | 0.504 | 0.579 | 0.339 | 0.635 | 0.599 | 0.581 | 0.315 | 0.693 | 0.441 | 0.547 |
| | LLaVA-Critic-7B (v0.5) | 0.575 | 0.677 | 0.73 | 0.556 | 0.427 | 0.521 | 0.537 | 0.366 | 0.568 | 0.62 | 0.571 | 0.353 | 0.703 | 0.435 | 0.546 |
| | LLaVA-Critic-7B | 0.593 | 0.687 | 0.707 | 0.587 | 0.432 | 0.544 | 0.564 | 0.338 | 0.596 | 0.628 | 0.591 | 0.37 | 0.686 | 0.464 | 0.556 |
| | LLaVA-Critic-72B | 0.587 | 0.672 | 0.86 | 0.588 | 0.475 | 0.536 | 0.618 | 0.366 | 0.628 | 0.608 | 0.568 | 0.39 | 0.721 | 0.473 | 0.578 |
| | GPT-4V* | 0.804 | 0.870 | 0.922 | 0.807 | 0.801 | 0.805 | 0.734 | 0.849 | 0.761 | 0.703 | 0.699 | 0.647 | 0.755 | 0.659 | 0.773 |
| | Gemini-Pro* | 0.717 | 0.840 | - | 0.770 | 0.678 | 0.793 | 0.688 | 0.658 | 0.711 | 0.652 | 0.471 | 0.358 | 0.265 | 0.400 | 0.615 |
| | LLaVA-1.6-34b* | 0.607 | 0.824 | 0.855 | 0.402 | 0.587 | 0.750 | 0.758 | 0.381 | 0.503 | 0.564 | 0.712 | 0.679 | 0.694 | 0.762 | 0.648 |
| | GPT-4o† | 0.774 | 0.776 | 0.934 | 0.835 | 0.628 | 0.618 | 0.737 | 0.513 | 0.741 | 0.770 | 0.706 | 0.722 | 0.887 | 0.660 | 0.736 |
| | GPT-4V† | 0.729 | 0.772 | 0.884 | 0.853 | 0.665 | 0.661 | 0.760 | 0.495 | 0.785 | 0.707 | 0.697 | 0.639 | 0.741 | 0.654 | 0.717 |
| Pair w.o. Tie (↑) | LLaVA-v1.5-7B | 0.617 | 0.571 | 0.637 | 0.598 | 0.411 | 0.544 | 0.452 | 0.554 | 0.653 | 0.562 | 0.672 | 0.600 | 0.558 | 0.631 | 0.576 |
| | LLaVA-NeXT (LLaMA-8B) | 0.565 | 0.684 | 0.473 | 0.526 | 0.460 | 0.526 | 0.516 | 0.549 | 0.634 | 0.592 | 0.641 | 0.648 | 0.673 | 0.716 | 0.586 |
| | LLaVA-OV-7B | 0.462 | 0.562 | 0.588 | 0.530 | 0.434 | 0.473 | 0.400 | 0.543 | 0.563 | 0.527 | 0.639 | 0.670 | 0.633 | 0.679 | 0.550 |
| | LLaVA-OV-72B | 0.691 | 0.780 | 0.811 | 0.714 | 0.623 | 0.634 | 0.625 | 0.57 | 0.737 | 0.685 | 0.755 | 0.702 | 0.746 | 0.736 | 0.701 |
| | LLaVA-Critic (LLaVA-v1.5-7B) | 0.732 | 0.757 | 0.665 | 0.659 | 0.574 | 0.623 | 0.650 | 0.544 | 0.710 | 0.719 | 0.649 | 0.589 | 0.708 | 0.602 | 0.656 |
| | LLaVA-Critic (LLaVA-NeXT) | 0.763 | 0.775 | 0.762 | 0.720 | 0.599 | 0.606 | 0.705 | 0.491 | 0.756 | 0.716 | 0.682 | 0.598 | 0.725 | 0.577 | 0.677 |
| | LLaVA-Critic-7B (v0.5) | 0.747 | 0.758 | 0.771 | 0.716 | 0.580 | 0.625 | 0.661 | 0.525 | 0.692 | 0.729 | 0.697 | 0.632 | 0.728 | 0.585 | 0.675 |
| | LLaVA-Critic-7B | 0.771 | 0.774 | 0.755 | 0.758 | 0.596 | 0.658 | 0.680 | 0.488 | 0.727 | 0.742 | 0.692 | 0.658 | 0.715 | 0.635 | 0.689 |
| | LLaVA-Critic-72B | 0.762 | 0.762 | 0.904 | 0.755 | 0.637 | 0.648 | 0.763 | 0.528 | 0.769 | 0.718 | 0.693 | 0.708 | 0.742 | 0.624 | 0.715 |

Table 8: Comprehensive results on MLLM-as-a-Judge. *: the results of GPT-4V, Gemini-pro, and the best open-source LMM as reported in the original paper (Chen et al., 2024); †: results from our evaluation of GPT-4V/4o using its original codebase.

| Method | LLaVA-W | L-Wilder | WildVision | LiveBench | VideoDC | MMHal | $MME^P$ | $MME^C$ | MMB-en | MM-Vet | MMStar |
|---|---|---|---|---|---|---|---|---|---|---|---|
| GPT-4V | 98.0 | 81.0 | 79.8 | 73.7 | 4.00 | 3.83 | 1409.4 | 517.1 | 75.0 | 49.9 | 57.1 |
| LLaVA-OV-7B | 90.7 | 67.8 | 54.0 | 77.1 | 3.75 | 3.19 | **1580.4** | 418.2 | **80.8** | **57.5** | 61.7 |
| + LLaVA-Critic-7B iter-1 | 96.7 | 70.6 | 60.5 | 81.2 | 3.77 | 3.62 | 1561.8 | **420.7** | **80.8** | 54.5 | 62.1 |
| + LLaVA-Critic-7B iter-2 | 97.0 | **72.2** | 65.2 | 83.9 | 3.82 | 3.67 | 1565.5 | 415.4 | 80.7 | 54.6 | 62.1 |
| + LLaVA-Critic-7B iter-3 | **100.3** | 71.6 | **67.3** | **84.5** | **3.87** | **3.91** | 1555.3 | 414.6 | 80.3 | 54.4 | **62.3** |
| LLaVA-OV-72B | 93.5 | 72.0 | 51.7 | 81.5 | 3.60 | 3.61 | 1683.2 | 578.9 | **85.9** | 63.7 | 66.1 |
| + LLaVA-Critic-72B iter-1 | 99.3 | 75.3 | 65.7 | 86.4 | 3.83 | 3.75 | 1683.2 | 584.3 | 85.6 | 67.0 | 66.4 |
| + LLaVA-Critic-72B iter-2 | 104.1 | 75.6 | 68.4 | 86.6 | **3.86** | 3.75 | 1681.1 | **586.4** | 85.7 | 66.7 | **66.5** |
| + LLaVA-Critic-72B iter-3 | **104.4** | **75.9** | **70.0** | **88.5** | **3.86** | 3.77 | **1686.1** | **586.4** | 85.4 | **67.1** | 66.4 |

Table 9: Performance of difference rounds of iterative DPO on LLaVA-OneVision. With the high-quality feedback from LLaVA-Critic, both LLaVA-OneVision 7B and 72B learn to refine its self-generated responses in a progressive manner, leading to overall better performance across various open-ended multimodal benchmarks.

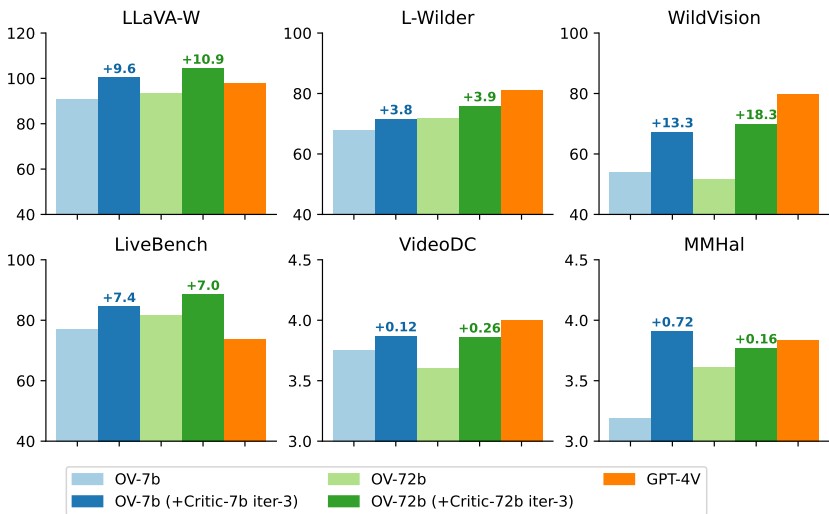

Figure 3: Visualization on the performance gain of LLaVA-OneVision obtained through LLaVA-Critic across 6 open-ended multimodal benchmarks.

## C  MORE QUALITATIVE COMPARISONS

We provide detailed examples of LLaVA-Critic in serving as a generalist evaluator for LMM responses, and improving visual chat through preference alignment.

- Table 10 shows an example of LLaVA-Critic ranking a pair of LMM responses. By accurately recognizing the visual content of the input image and grounding the differences between the responses, LLaVA-Critic offers judgments consistent with human evaluators, along with clear justifications.

- Table 11 presents an example of LLaVA-Critic scoring an LMM response using a reference answer from GPT-4. LLaVA-Critic closely follows the evaluation prompt and, by referring to the image content, accurately identifies the strengths and weaknesses of the response at both overall and fine-grained levels. Consequently, it delivers a fair evaluation score and detailed justification aligned with GPT-4o.

- Table 12 illustrates LLaVA-Critic's effectiveness in enhancing LLaVA-OneVision's visual chat performance through iterative DPO training. By progressively aligning with LLaVA-Critic's preferences, LLaVA-OneVision learns to generate responses that are more structured, detailed and better tailored to user needs.

**Example 1: LLaVA-Critic in Judging LMM Responses (Pairwise)**

| | |
|---|---|
| 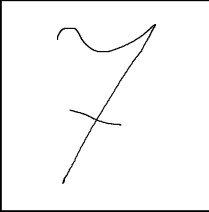 Critic Input | **Question:** What this image presents?
**Response 1 (model_a):** The image is a black and white sketch of a line that appears to be in the shape of a cross. The line is a simple and straightforward representation of the cross shape, with two straight lines intersecting at a point.
**Response 2 (model_b):** This is a handwritten number seven.
**System**: Given an image and a corresponding question, please serve as an unbiased and fair judge to evaluate the quality of the answers provided by a Large Multimodal Model (LMM). Determine which answer is better and explain your reasoning with specific details. Your task is provided as follows: Question: []\n Response 1: []\n Response 2: []\n ASSISTANT: |
| Human | The winner is Response 2. |
| LLaVA-Critic-7B | Response 2 is better than Response 1 because it accurately identifies the image as a handwritten number seven. Response 1 incorrectly describes the image as a black and white sketch of a line in the shape of a cross, which does not match the visual content of the image. Response 2 correctly interprets the image, providing a precise and accurate description that aligns with the actual content of the image. This makes Response 2 more helpful and accurate for understanding what the image presents. |
| LLaVA-OV-7B | Response 1 is better because it accurately describes the image as a simple and straightforward representation of a cross shape, which is what the image actually depicts. Response 2, while also describing the image, does not accurately convey the intended meaning of the image, as it suggests that the image is a handwritten number seven, which is not the case. |

Table 10: Example of LLaVA-Critic in ranking LMM response pairs. While LLaVA-OV misinterprets the handwritten number and gives inaccurate judgment, LLaVA-Critic aligns with human assessments and offers clear, visually grounded justifications.

**Example 2: LLaVA-Critic in Judging LMM Responses (Pointwise)**

| | |
|---|---|
| Critic Input | 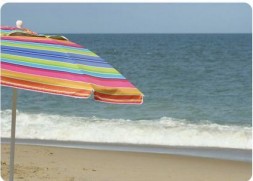 **Question:** Can you describe the weather conditions and overall atmosphere of the scene depicted in the image? **Assistant 1 (GPT-4):** The image depicts a calm and serene beach scene. The sky appears to be clear with no signs of clouds, indicating good weather conditions. The calmness of the ocean, suggested by the gentle waves crashing on the shore, further enhances the tranquility. The presence of a colorful beach umbrella provides a vibrant contrast to the more subdued hues of the sea and sand, implying a relaxed and leisurely atmosphere commonly associated with a sunny day at the beach. **Assistant 2 (Evaluated LMM response):** The image depicts a sunny and pleasant day at the beach, with a colorful umbrella providing shade on the sandy shore. The presence of the umbrella suggests that it is likely a hot day, and people might be seeking relief from the sun by sitting under the umbrella or going for a swim in the ocean. The atmosphere is likely relaxed and enjoyable, with people spending time at the beach to unwind and take in the beautiful surroundings. The sunny weather and the calming sound of the waves further contribute to the overall pleasant and leisurely vibe of the scene. **System:** We would like to request your feedback on the performance of two AI assistants in response to the user question displayed above. The user asks the question on observing an image shown to you. Please rate the helpfulness, relevance, accuracy, level of details of their responses. Each assistant receives an overall score on a scale of 1 to 10, where a higher score indicates better overall performance. Assume assistant 1 always receive a score of 10 and is the correct answer. Please first output a single line containing only two values indicating the scores for Assistant 1 and 2, respectively. The two scores are separated by a space. In the subsequent line, please provide a comprehensive explanation of your evaluation, avoiding any potential bias and ensuring that the order in which the responses were presented does not affect your judgment. |
| GPT-4o | 10 9 Assistant 1 provided a detailed and accurate description of the weather conditions and overall atmosphere of the scene depicted in the image. The response highlighted the clear sky, calm ocean, and the presence of a colorful beach umbrella, which all contribute to a serene and tranquil atmosphere. The description was comprehensive and captured the essence of the scene effectively. Assistant 2 also provided a good description, noting the sunny and pleasant weather, the colorful umbrella, and the relaxed atmosphere. However, the response included some assumptions, such as people seeking relief from the sun and going for a swim, which were not directly observable in the image. This slightly reduced the accuracy and relevance of the response. |
| LLaVA-Critic-7B | 10 9 Assistant 1 provided a detailed and accurate description of the weather conditions and overall atmosphere depicted in the image. The response highlighted the clear sky, calm ocean, and the presence of a colorful beach umbrella, which all contribute to a serene and tranquil beach scene. The description was relevant, accurate, and included a good level of detail. Assistant 2 also gave a good description, mentioning the sunny day, the colorful umbrella, and the relaxed atmosphere. However, it made an assumption about people seeking relief from the sun and going for a swim, which is not directly supported by the image. This slightly detracts from the accuracy and relevance of the response. Therefore, Assistant 2 receives a slightly lower score. |
| LLaVA-OV-7B | 10 9 Assistant 1 provided a clear and detailed description of the image, including the weather conditions and overall atmosphere. The response was relevant to the question asked and accurately described the scene depicted in the image. The level of detail was appropriate for the task, providing enough information for the user to understand the context of the image. Assistant 2 also provided a clear and detailed description of the image, but it was slightly less concise than Assistant 1's response. While the response was still relevant to the question asked and accurately described the scene depicted in the image, it did not provide as much detail as Assistant 1's response. However, the level of detail provided by Assistant 2 was still sufficient for the task, and the response was still helpful and accurate. Overall, both assistants performed well in their responses, but Assistant 1 provided a slightly more detailed and comprehensive response, which is why it receives a score of 10, while Assistant 2 receives a score of 9. |

Table 11: Example of LLaVA-Critic conducting pointwise scoring on an LMM response for LLaVA-Wilder. Unlike LLaVA-OV, which offers vague and uninformative explanations, LLaVA-Critic delivers a thorough and fair assessment with clear justifications on both the strengths and weaknesses of the evaluated response, closely resembling the depth and fairness of GPT-4o's assessments.

**Example 3: LLaVA-Critic Enhances LLaVA-OneVision in Visual Chat**

| Question | Discuss how this creative twist on a classic work of art might be interpreted differently by various audiences. |
|---|---|
| LLaVA-OV-72B | Different audiences might interpret this creative twist on the Mona Lisa in various ways. Art enthusiasts might appreciate the skillful execution and the playful reinterpretation of a classic, seeing it as a commentary on the universality of beauty and expression across species. Dog lovers could find joy in seeing their favorite breed portrayed with such dignity and elegance. Some might view it as a humorous take on high art, making it more accessible and relatable. Others might see it as a statement on the anthropomorphism of pets, reflecting how we often attribute human-like qualities to animals. Overall, the varied interpretations highlight the versatility of art in engaging diverse perspectives and emotions. |
| LLaVA-OV-72B + Critic-72b iter-3 | This creative twist on the Mona Lisa, featuring a dog's face, can be interpreted in various ways by different audiences. |
| | For art enthusiasts, it might be seen as a playful commentary on the nature of portraiture and the human tendency to anthropomorphize animals. It challenges traditional notions of beauty and subjectivity in art, inviting viewers to reconsider what makes a portrait compelling. |
| | For dog lovers and pet owners, the image could evoke a sense of joy and familiarity, as they see their beloved pets reflected in one of the most iconic artworks in history. It might also highlight the deep bond between humans and their pets, suggesting that animals share in our cultural heritage. |
| | In a broader social context, this image could be interpreted as a reflection of contemporary society's obsession with internet culture and memes. The substitution of the Mona Lisa's face with a dog's could be seen as a satirical take on how viral content often overshadows classical art in modern media. |
| | Academically, this piece could spark discussions about copyright, artistic originality, and the public domain. It raises questions about the boundaries between homage and appropriation in art, encouraging critical thinking about intellectual property rights and creative freedom. |
| | Overall, the diverse interpretations of this image underscore the power of art to engage, provoke, and connect people across different backgrounds and perspectives. |

Table 12: Example of LLaVA-Critic in preference learning. Leveraging reward signals from LLaVA-Critic, LLaVA-OneVision improves its visual chat ability through iterative DPO, delivering more detailed, valuable, and structured point-by-point responses.

