# OpenReview forum: "LLaVA-Critic: Learning to Evaluate Multimodal Models"
_ICLR.cc/2025/Conference — ICLR 2025 Conference Withdrawn Submission_

### Official Review · Reviewer_9fcY · 2024-10-28

**Soundness:** 3
**Presentation:** 2
**Contribution:** 2
**Rating:** 3
**Confidence:** 3

**Summary:**

This paper introduces a new open-source multimodal large language model, LLaVA-Critic, which is fine-tuned on a high-quality evaluation dataset constructed with GPT-4 assistance. This model enhances the ability to assess various tasks across different multimodal scenarios. Two evaluation metrics are used: pointwise scoring and pairwise ranking, which can be applied in two ways—as a judgement or as a preference reward generator. Experiments demonstrate that LLaVA-Critic surpasses the original model in evaluation capabilities and effectively guides preference learning.

**Strengths:**

Enhancing large models' evaluation capabilities in multimodal scenarios is the central focus of this paper. This paper improves the evaluation ability of an existing large model by fine-tuning it on a high-quality evaluation dataset, resulting in a new multimodal large language model. The paper is well-written, with smooth flow and well-supported arguments, making it easy to follow.

**Weaknesses:**

However, this paper does not have enough innovation. It only opens up the relevant datasets and uses them to finetune a new multimodal large model, and there is no new methodological innovation. The experimental section is also lacking in comprehensive baselines. The differences in pointwise scoring and pairwise ranking capabilities between LLaVA-Critic and existing multimodal large language models are not compared, as LLaVA itself is not currently the most capable multimodal model. Further experiments are needed to determine whether LLaVA-Critic retains an advantage in task evaluation compared to more advanced models like Qwen2-VL and InternVL. Additionally, the paper lacks a detailed exploration of data requirements for LLaVA-Critic; it appears that 50% of the data size can still train an effective MLLM.

**Questions:**

Firstly, I think more comparative experiments are needed to determine if LLaVA-Critic can still maintain a leading position when compared to current advanced MLLMs. Secondly, I am curious about LLaVA-Critic's requirements for training data scale. Finally, I wonder if the fine-tuning process affects LLaVA-Critic's original question-answering ability, as overfitting to the current dataset is inevitable with fine-tuning. Does this impact its performance on the original tasks, such as potentially improving evaluation capabilities but compromising original image-question answering?

---

### Official Review · Reviewer_pH5G · 2024-10-29

**Soundness:** 2
**Presentation:** 3
**Contribution:** 2
**Rating:** 6
**Confidence:** 3

**Summary:**

This paper presents LLaVA-Critic, a multi-modal model that evaluates the response quality of general queries. LLaVA-Critic is trained using a dataset of examples of evaluating responses given some queries. There are two types of examples:

1) point-wise scoring: the example has a query, a response, then an evaluation prompt giving a rubric, usually from the range 0 to 100
2) pair-wise scoring: the example has a query, two responses, and then an evaluation prompt to have the LM pick a better response.

The dataset prompts come from many sources, and the dataset responses were generated using GPT-4o.

The authors then evaluated LLaVA-Critic on a wide range of multi-modal evaluation tasks and found LLaVA-Critic to perform better than baselines. Furthermore, an iterative DPO model is trained based on the LLM-as-a-judge feedback from LLaVA-Critic, and the DPO models perform better than the SFT baseline in a series of benchmarks.

**Strengths:**

The results seem strong: good performance on a series of benchmarks. The paper is written clearly, explaining the data collection process well.

**Weaknesses:**

The paper could benefit from some ablation studies to offer insights. For example, it was mentioned "We randomly select 20k pairs where the average score gap between responses is greater than 0.6. Besides, to ensure diversity in the preferences, we randomly sample 5k pairs where the two responses had identical scores across all three dimensions to serve as “Tie” training data." Why was this decision made? What are their effects on the final model artifacts?

Table 4 and Table 5 might benefit from including other open-source multi-modal models like

https://huggingface.co/mistralai/Pixtral-12B-2409
https://huggingface.co/meta-llama/Llama-3.2-3B-Instruct

**Questions:**

Could you explain more on how you decided to curate these data? Why did you decide on `LLaVA-in-the-Wild`, `LLaVA-Wilder` and others?

What was the iteration process? I imagined prob dozens of models were trained for the paper and only the best one was presented. What makes the best one stand out?

---

### Official Review · Reviewer_W2ty · 2024-11-04

**Soundness:** 3
**Presentation:** 3
**Contribution:** 2
**Rating:** 5
**Confidence:** 4

**Summary:**

The paper introduce LLaVA-Critic, an open source LMM that has been trained as an evaluator rather than a generator of content. To do so, the paper uses a variety of datasets that include prompt (image+question), responses, and their evaluation. Throughout most of these datasets, the evaluations are done be GPT-4o, and LLaVA-Critic is trained to imitate these evaluations through a standard cross-entropy loss. The model is trained both for point wise evaluation (i.e. LLM-as-a-Judge), where the model needs to output a score between 1 and 10, and pairwise comparisons, where the model has to output a preference between two candidate responses. The experimental results focus on these two settings as well, where results show that the proposed model closely matches the performance of GPT-4o for both settings, and that it performs better than open-source alternatives within a DPO algorithm pipeline.

**Strengths:**

The presentation of the paper is good, with authors being clear and detailed about the procedure being followed. For example, the references to previous datasets, prompts are either well cited in the main paper or detailed in the appendix. Figure 1 also helps understanding what constitutes the majority of the training dataset. The paper mentions that their method is open-source, which indicates that the weights will be released - a net positive for the community. Most of the claims in the empirical section are appropriate with respect to the obtained results, which indicates a fair evaluation from the authors.

**Weaknesses:**

The strongest concern about the paper is that there is nothing that is particularly surprising, or insightful. In essence, the procedure describes entails to distilling GPT-4o into an open-source model using standard cross-entropy, and showing performance that is close to that of GPT-4o. Although this is great for the community, I am struggling to see what the contribution is from a scientific perspective. I can see how this paper would be valuable within a Datasets & Benchmarks track as the one available in NeurIPS, which would likely be a better fit. For a scientific paper, I think there is a necessity to try to answer some question, improve some empirical understanding, or present a surprising phenomenon. At the moment, everything is very as-expected and very standard, ressembling a technical report.

For the preference learning experiments, why isn't there a comparison with GPT-4o directly expressing preferences to create a reward model for DPO? This seems like the more appropriate comparison rather than using GPT-4V for preference alignment. Also, the authors claim that their approach is superior to other competing methods, but isn't this expected given that the other approaches do not fine-tuning from a teacher (i.e. GPT-4o preferences)? Aren't most of the other baselines simply complimentary to fine-tuning on a teacher?

**Questions:**

For the MLLM-as-a-Judge benchmark, how much overlap is there between that benchmark and the ones used to train the model? These experiments are referred to as "out-of-domain", but this is not clearly shown in the paper.

In Table 1, "dominant" might be a typo.

The references for generating reward signals in RL could be better chosen, especially as o1 is a closed source model.

---

### Official Review · Reviewer_i5xB · 2024-11-12

**Soundness:** 2
**Presentation:** 2
**Contribution:** 2
**Rating:** 5
**Confidence:** 3

**Summary:**

This paper proposes a high-quality dataset for evaluation purposes. Based on this dataset, it trains an MLLM mainly for evaluation use. The results show that LLAVA-critic achieved SOTA performance in the evaluation tasks.

**Strengths:**

1. Taking LMM as a critic is a very important research direction. It can be used to further fine-tune the LMM.
2. The authors open source the critic instruction data, codebase, and model checkpoints, which is a big contribution to the community.

**Weaknesses:**

From the methodology perspective, I do not get much insight. It seems this is engineering work, gathering the large-scale dataset and training a large model with lots of resources. I would give an accept if this is a benchmark setting. But for the main track, the novelty is not sufficient.

**Questions:**

My main concern is for the novelty side.

---

### Note · Authors · 2024-11-13

I have read and agree with the venue's withdrawal policy on behalf of myself and my co-authors.